# The role of morphology on the spatial distribution of short-duration rainfall extremes in Italy

Paola Mazzoglio[1], Ilaria Butera[1], Massimiliano Alvioli[2], Pierluigi Claps[1]

[1]Department of Environment, Land and Infrastructure Engineering, Politecnico di Torino, Torino, 10129, Italy
[2]Istituto di Ricerca per la Protezione Idrogeologica, Consiglio Nazionale delle Ricerche (CNR), Perugia, 06128, Italy

*Correspondence to*: Paola Mazzoglio (paola.mazzoglio@polito.it)

**Abstract.** The dependence of rainfall on elevation has frequently been documented in the scientific literature and may be relevant in Italy, due to the high degree of geographical and morphological heterogeneity of the country. However, a detailed analysis of the spatial variability of short-duration annual maximum rainfall depths and their connection to the landforms does
not exist. Using a new, comprehensive and position-corrected rainfall extreme dataset ($I^2$-RED), we present a systematic study of the relationship between geomorphological forms and the average annual maxima (index rainfall) across the whole of Italy. We first investigated the dependence of sub-daily rainfall depths on elevation and other landscape indices through univariate and multivariate linear regressions. The results of the national-scale regression analysis did not confirm the assumption of elevation being the sole driver of the variability of the index rainfall. The inclusion of longitude, latitude, distance from the
coastline, morphological obstructions and mean annual rainfall contributes to explain a larger percentage of the variance, even though in different ways for different durations (1- to 24-hours). After analyzing the spatial variability of the regression residuals, we repeated the analysis on geomorphological subdivisions of Italy. Comparing the results of the best multivariate regression models with univariate regressions applied to small areas, deriving from morphological subdivisions, we found that "local" rainfall-topography relationships outperformed the country-wide multiple regressions, offered a uniform error spatial
distribution and allowed to better reproduce the effect of morphology on rainfall extremes.

## 1 Introduction and background

The spatial patterns of rainfall depth statistics are known to be affected by the geomorphological setting (Smith, 1979; Basist et al., 1994; Prudhomme and Reed, 1998; Prudhomme and Reed, 1999; Faulkner and Prudhomme, 1998). The impact of orography on daily, multi-daily and annual precipitation events can generally be attributed to the so-called "orographic
enhancement of precipitation", i.e. an increase in rainfall depth along the windward slope of a relief and a decrease on the lee side, due to the lifting and the consequent drying of the air mass (Smith, 1979; Daly et al., 1994; Frei and Schär, 1998; Napoli et al., 2019). In complex landscape, this effect can entail significant precipitation values also on the lee side, due to landforms that causes delay in the hydrometeorological formation of precipitation and falling raindrops (Smith, 1979).

The impact of the orography on extreme rainfall depths and the complicated atmosphere-orography interactions for large areas
are still not sufficiently understood for sub-daily rainfall events. In a country like Italy, characterized by a high degree of morphological heterogeneity (Figure 1) these relations assume an evident importance, considering the significant exposure to Mediterranean storms (Claps and Siccardi, 2000). The focus of this study is the entire Italian territory ($\approx$300,000 km$^2$) considered as a representative case, both in terms of variety of landforms and in terms of variability of the rainfall extremes, as will be seen in the following.

Most of the existing studies in Italy have focused on limited areas (Allamano et al., 2009; Caracciolo et al., 2012; Pelosi and Furcolo, 2015; Furcolo et al., 2016; Furcolo and Pelosi, 2018; Libertino et al., 2018; Formetta et al., 2022) and the only attempt to deal with sub-daily data covering the entire nation was made by Avanzi et al. (2015).

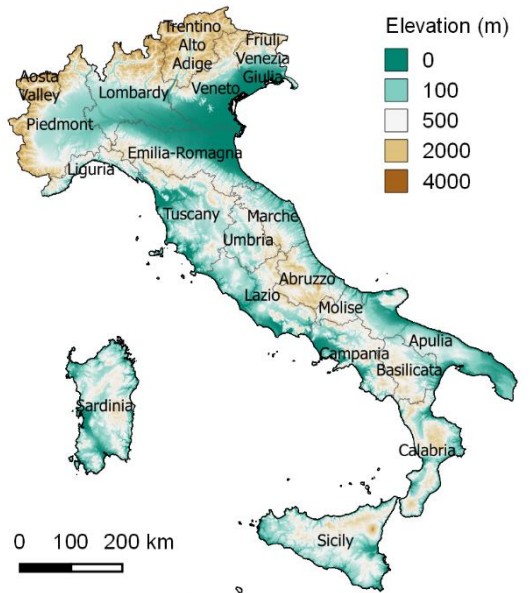

**Figure 1. Elevation data with the boundaries of the 20 Italian administrative regions. Source: Shuttle Radar Topography Mission (Farr et al. (2007)).**

These studies suffered from the lack of a comprehensive and quality-assessed national database for sub-daily extremes. Several of them analyzed the Italian Alpine area. For instance, Frei and Schär (1998) focused on the entire European Alps region, and showed that foothills enhance monthly and seasonal precipitations, while inner valleys produce an orographic shielding effect on rainfall. Nevertheless, they did not find a unique precipitation depth–elevation relationship that could be considered valid for the entire Alps and attributed the observed variability to the effects of slope and shielding. Allamano et al. (2009) investigated the dependence of sub-daily annual maximum rainfall depths on elevation over the Italian Alpine region. They found a significant decreasing trend for increasing elevations and a non-uniform slope coefficient over the longitude range. The slope of the rainfall depth – elevation regression was shown to decrease for event durations that increased from 1 to 24 hours. Libertino et al. (2018) showed, in the western sector of the Italian Alps (Figure 1), that shorter durations (1 – 3 hours) are characterized by a negative slope coefficient as a function of elevation (statistically significant at a 5% level), while longer durations (12 – 24 hours) show a positive slope and also a significant correlation, while the trend of the extremes over 6-hours loses significance with the elevation. Formetta et al. (2022) identified over Trento province a reverse orographic effect for hourly and sub-hourly durations and an orographic enhancement for duration of about 8 hours (or longer).

Other regional works that attempted to identify orographic effects in the Mediterranean part of Italy are available for Campania and Sicily. Pelosi and Furcolo (2015) and Furcolo et al. (2016) analyzed the daily annual maximum rainfall depths over Campania (see Figure 1 for the geographical location) and attempted to explain systematic variations as being the result of the presence of orographic barriers, identified through the application of an automatic geomorphological procedure (Cuomo et al., 2011). Their results showed a link between orographic elements and a local increase in rainfall depths, and allowed orographic elements that produced enhanced variability of extreme rainfall to be identified. The same group later worked on sub-daily annual maximum rainfall depths (Furcolo and Pelosi, 2018) and proposed a power-law amplification factor of rainfall over three mountainous systems.

Caracciolo et al. (2012) found, in Sicily, that the longitude, latitude, distance from the sea and a concavity index are the variables that govern the spatial variability of rainfall depths. However, these authors found that no linear relationship between sub-daily annual maximum rainfall depths and elevation was significant at a 5% level over the entire island of Sicily.

All of the previously mentioned analyses refer to an analytic relationship that connects annual maximum rainfall depths of various durations, i.e. the Average Depth-Duration (ADD) curve of the simple-scaling approach, which is usually represented by a power-law:

$$\bar{h}_d = a \cdot d^n \qquad\qquad (1)$$

where $\bar{h}_d$ is the average of the annual maximum rainfall depths of duration $d$, $a$ is a scale factor and $n$ is a scaling exponent.

Coefficient $a$ represents the best unbiased linear estimation of the 1-hour average rainfall depths, considering that $\bar{h}_1 \cong a$. Avanzi et al. (2015) analyzed the spatial variability of the ADD curve parameters, $a$ and $n$, at a national scale, as obtained from measurements of 1,494 stations distributed throughout Italy. They referred to the so-called "reverse orographic effect", i.e. the relationship found between parameter $a$ and elevation, which shows a decreasing trend. On the other hand, the scaling exponent $n$ appears to increase non-linearly with the elevation. More details are provided in the following section.

On the basis of the described background, and on the significant improvements offered by a new, up-to-date rainfall dataset, i.e. the Improved Italian – Rainfall Extreme Dataset or I²-RED (Mazzoglio et al., 2020), the present study has considered more than 3,700 stations with at least 10 years of data to relate the average rainfall depths in all the durations (index rainfall) to several morphological variables, and investigate their dependency on elevation and on other geomorphological and climatological parameters.

Searching for models that allow the index rainfall to be estimated for various durations in any location in Italy is the first, important, necessary step toward addressing the building of Depth-Duration-Frequency curves over the entire country. For this purpose, simple (Sect. 2) and multiple (Sect. 3 and 4) national scale regression models were first investigated. We then introduced four geomorphological classifications to perform local-scale regression analysis in order to tackle the evident spatial clustering of the regression residuals (Sect. 5). The comparisons made between the results obtained from the wide-area and

the local regressions allowed the role of the morphology on rainfall variability to be discussed, as shown in Sect. 6. Some conclusions are drawn in Sect. 7.

## 2 National-scale simple regression analysis

### 2.1 Methods

As the first step of the analysis, we investigated the influence of elevation on the spatial distribution of the average of annual

maximum rainfall depths. We calculated the ADD curve parameters for all the stations of the I²-RED dataset (Mazzoglio et al., 2020) to compare them with previous studies (mainly Avanzi et al., 2015). Parameters $a$ and $n$ (cf. Eq. (1)) were initially obtained by means of a linear regression of the logarithm of the average of all the available extremes over the 1- to 24-hour durations. We computed the median values of these parameters, for all over Italy, to compare them with those of Avanzi et al. (2015), who grouped the stations into elevation ranges of 50 meters up to 1,000 m a.s.l., and into intervals of 100 meters for

higher elevations. We then plotted both series of medians ($a$ and $n$) against the median elevation of each interval (to consider that the distribution of the rain gauges in each elevation interval was skewed) and fitted regression models.

We studied the differences between the measured and estimated rainfall statistics to assess the effectiveness of the regression models, considering the observed averages of the extremes over 1, 3, 6, 12 and 24 hours. We obtained performance indices for each station using the estimation errors $\Delta_d$:

$$\Delta_d = h_{avg}(d) - \hat{a} \cdot d^{\hat{n}} \, , \qquad\qquad\qquad (2)$$

where $h_{avg}(d)$ is the sample average of the extreme rainfall depth for duration $d$, while $\hat{a}$ and $\hat{n}$ are the estimates of parameters $a$ and $n$.

In this paper, we show and discuss only the results related to the shortest and the longest of the five durations (1 and 24 hours), as they can be considered the most representative of the different classes of rainfall events (convective and stratiform,

respectively). The corresponding dependent variables are called $\bar{h}_1$ and $\bar{h}_{24}$ in the following.

The error statistics that were computed are the bias, the mean absolute error (*MAE*), the root mean square error (*RMSE*) and the Nash-Sutcliffe model efficiency (*NSE*) coefficient (Nash and Sutcliffe, 1970; Wasserman, 2004). Among all the statistics, particular attention was dedicated to spatial bias, i.e. the bias evaluated as the difference between the spatial mean of the observations over a generic area, and the corresponding values predicted by the model.

## 2.2 Results

By applying the procedure described in Sect. 2.1, we obtained results that are in agreement with those of Avanzi et al. (2015), that is:

1) Parameter $a$ decreases linearly with the elevation ($R^2 = 0.89$), through the equation:

$$a = 30.61 - 0.0060 \cdot z , \tag{3}$$

which is comparable with the equation obtained in Avanzi et al. (2015):

$$a = 29.17 - 0.0062 \cdot z . \tag{4}$$

2) Parameter $n$ increases non-linearly with the elevation ($R^2 = 0.86$), through the equation:

$$n = 0.54 - exp[-0.000077 \cdot (z + 1650)] . \tag{5}$$

For comparison purposes, Avanzi et al. (2015) obtained

$$n = 0.54 - exp[-0.00086 \cdot (z + 1452)] \tag{6}$$

for the latter parameter, with only a slight difference in $R^2$ (0.89).

The fitting of the four models is reported in Supplement n°1.

As already mentioned, parameter $a$ is roughly equivalent to $\bar{h}_1$. Its overall inverse dependence on elevation is somewhat counter-intuitive, even though other authors have confirmed this dependence (e.g. Allamano et al., 2009; Marra et al., 2021).

The error statistics computed on the two sets of residuals, in this work and in that of Avanzi et al. (2015), are listed in Table 1. The results show that the increase in the number of stations and the recording length achieved in I²-RED have led to an improvement compared to the results of Avanzi et al. (2015). This result is not surprising, but more insights can be derived from the observation of the spatial distribution of the residuals, which were not discussed explicitly in the previous literature. In this regard, we mapped differences $\Delta_1$ and $\Delta_{24}$ to investigate where the under- and over-estimations show spatial coherence. The maps, reported in Figure 2, clearly show that clusters of residuals with high residuals of the same sign emerge in various areas of the country: for instance, many coherent errors larger than 3 times the MAE are present in the Liguria region (see Figure 1 for the geographic position) for both durations. Therefore, despite the high $R^2$ values, significant spatially correlated errors can undermine the practical validity of these general relationships.

On the basis of these results, the need for a more detailed spatial analysis of these variables became evident. A set of new analyses, aimed at reducing the local bias and increasing the reliability of the results, was therefore introduced.

| Error statistic | $\bar{h}_1$ - Avanzi et al., 2015 (Eqs. (4) and (6)) | $\bar{h}_1$ - This paper (Eqs. (3) and (5)) | $\bar{h}_{24}$ - Avanzi et al., 2015 (Eqs. (4) and (6)) | $\bar{h}_{24}$ - This paper (Eqs. (3) and (5)) |
|---|---|---|---|---|
| Bias (mm) | 2.65 | 1.07 | 9.64 | 6.05 |
| MAE (mm) | 5.48 | 5.29 | 22.22 | 22.27 |
| RMSE (mm) | 7.39 | 6.98 | 32.81 | 31.99 |
| NSE (-) | -0.01 | 0.10 | -0.02 | 0.03 |

**Table 1. Comparison of national-scale error statistics related to the estimates performed with our data and those of Avanzi et al. (2015). The results were obtained with Eqs. (3) to (6).**

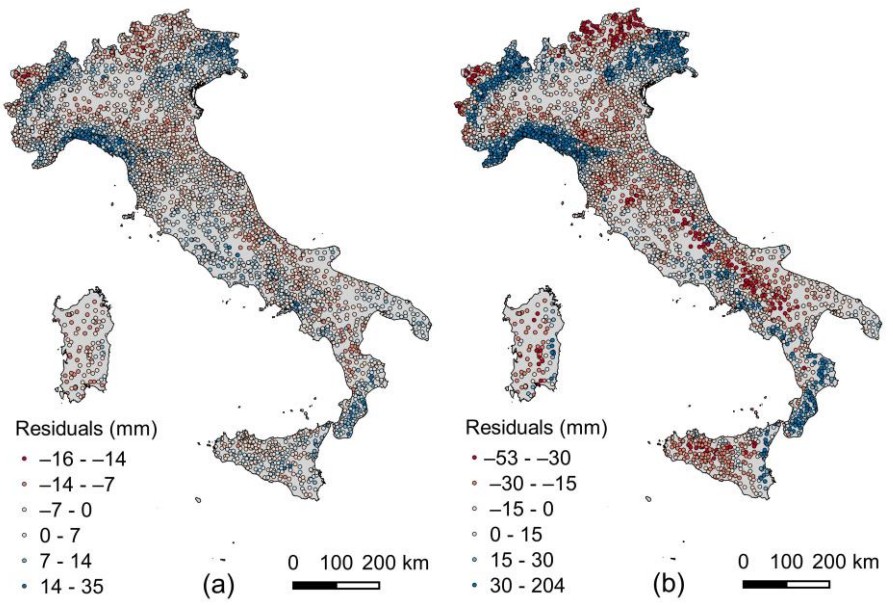


**Figure 2. Residuals of the estimations of the 1-hour (a) and 24-hour (b) durations performed using Eqs. (3) and (5).**

## 3 National-scale multiple regression analysis

### 3.1 Methods

In an attempt to improve the evaluation of the relationships between rainfall and topography, we undertook an analysis of the
relationships between rainfall and several geomorphological (and climatological) parameters, which may complement the explanatory power of elevation. Unlike what was done in Avanzi et al. (2015), multivariate models were used in the literature to relate rainfall statistics and various morphological variables, both of which were evaluated at the same location. In these approaches, no aggregated or median spatial statistics of rainfall were considered. Prudhomme and Reed (1998, 1999), for instance, identified meaningful geographic and morphological attributes of each location as good explanatory variables of the
daily rainfall maxima in Scotland. They showed that obstruction indices, derived from the orography, and the distance from the coastline, are able to define how morphological barriers influence the characteristics of the extremes. These appear to work better that the EXPO variable used by Basist et al. (1994) and Konrad (1996).

Basist et al. (1994) defined EXPO as the distance between a rain gauge and an upwind barrier whose elevation is at least 500 m higher than the station. Konrad (1996) suggested an elevation of the barrier at least 150 m higher than the station.
Prudhomme and Reed (1998) also tried to use this variable, setting the elevation difference at 200 m but concluded that the definition of EXPO has several drawbacks, as it is based on arbitrary thresholds and is defined assuming a specific and subjective direction.

Introducing new variables with omni-directional meaning, as the distance from the sea, the obstruction and the barrier, which are evaluated in the 8 main directions, Prudhomme and Reed (1998) were able to explain a much larger percentage of
variability in the annual maximum daily rainfall than that explained by the EXPO variable.

Caracciolo et al. (2012) applied this latter approach on the Sicily Island (South of Italy): they found that the longitude, elevation, a barrier obstruction index and the distance from the coastline are able to represent the spatial variability of parameter *a* for the whole island, while the longitude, elevation, a concavity index and the slope are able to satisfactorily describe the variability of exponent *n*. They also noticed that different descriptors became significant when analyzing smaller portions of
the island.

Based on the above considerations, in this work we followed the approach suggested by Prudhomme and Reed (1998, 1999), considering two groups of variables computed for each station location, i.e.:

a) geographic and climatic variables, which do not require computation and do not depend on the landscape forms, that is, longitude (*LON*, expressed in the WGS84 UTM32N reference system, in m), latitude (*LAT*, expressed in the WGS84 UTM32N reference system, in m), elevation above sea level (*z*, in m), minimum distance from the coastline (*C*, in km), and mean annual rainfall (*MAR*, taken from Braca et al., 2021, in mm); the latter represents a very robust climatological variable, which is seldom used as ancillary information but easily available throughout the world thanks to the presence of various rainfall databases (Schneider et al., 2011; Fick and Hijmans, 2017; Muñoz Sabater, 2019).

b) morphological variables, or descriptors, based on a Digital Elevation Model (DEM), computed for each cell in a square grid. These variables are:

- Slope (*S*, in degrees), defined as the angle of the inclination of the terrain to the horizontal, which is evaluated using the 8 closest DEM cells;
- Obstruction (*OBS*, in degrees), defined as the maximum angle needed to overcome the highest orographic obstacles in the eight main cardinal directions (i.e., the maximum of the angles subtended by the line that connects the rain gauge with the highest orographic peak within a 15 km radius in the eight main directions, see Figure 3);
- Barrier (*BAR*, in m), defined as the distance between the rain gauge and the highest orographic obstacle defined in OBS (Prudhomme and Reed, 1998 and 1999; see Figure 3);
- Maximum slope angle (*MSA*, in degrees), i.e. the angle with the greatest slope needed to overcome obstacles within a 15 km radius in the eight main directions (see Figure 3);
- Maximum slope angle distance (*MSAD*, in m), defined as the equivalent of *BAR*, but computed with respect to *MSA* (see Figure 3);
- Openness (*OP*, in radians), defined as a mean angular measurement of the relationships between the surface of the relief and the horizontal distances, in the eight main directions (Yokoyama et al., 2002).

The values of all of these variables depend on the landscape forms and can vary according to the resolution of the used DEM. In our case, after thorough consideration, we adopted the Shuttle Radar Topography Mission (SRTM) DEM, which has a resolution of 30 meters (Farr et al., 2007). However, the openness required to be evaluated on the SRTM DEM resampled at a resolution of 500 m due to computational limitations. This computation was conducted with the SAGA "Topographic openness" module, using a radial limit of 5 km. This value was obtained after testing different radial limits and selecting the one that presents the best correlation with the mean rainfall depth.

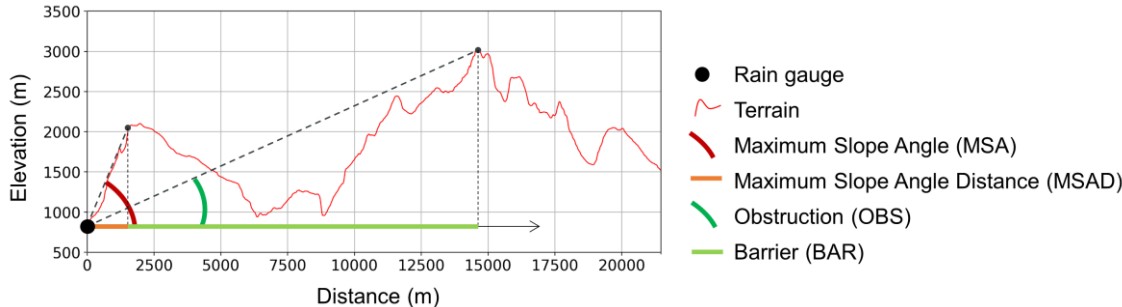

**Figure 3. Representation of the MSA, MSAD, OBS and BAR morphological variables.**

Multiple linear regression models were built, based on the relationship:

$$Y = \mathbf{X} \cdot \boldsymbol{\beta} + \boldsymbol{\delta} = \sum_{i=1}^{N} \beta_i X_i + \boldsymbol{\delta} \,, \tag{7}$$

where the dependent variable *Y* is related to the matrix of the independent variables *X*, or covariates. $\boldsymbol{\beta}$ in Eq. (7) is the vector that contains the regression model coefficients and $\boldsymbol{\delta}$ is the vector of the residuals.

In order to select the best model equation, the number *i* of covariates can be increased as necessary, according to the criteria of statistical significance of the estimated parameters. Caracciolo et al. (2012), for instance, used a stepwise regression approach. In this paper, we have preferred to use a generalized multiple regression approach whereby increasing the number of covariates to *i+1* does not necessarily preserve the descriptors that were the most significant at step *i*. This approach entails

always considering all the possible combinations of 2, 3 or 4 covariates until the "best" model is found. Other tests made using five or more variables did not lead to significantly higher $R^2_{adj}$ values.

The "best" regression model was selected on the basis of an analysis of the regression residuals, favoring models with the highest adjusted coefficient of determination, $R^2_{adj}$. Student's t-test was used to quantify the significance of the independent variables. We checked for the possible presence of multicollinearity for each model in which all the covariates passed Student's t-test, as this could lead to the formulation of an unstable model. Multicollinearity was measured using the Variance Inflation Factor ($VIF$), which is determined by placing the $j$-$th$ independent variable as the dependent variable and calculating the coefficient of determination $R^2_{adj}$ of the multiple regression performed on the remaining $p - 1$ independent variables (Eq. (8)).

$$VIF_j = \frac{1}{1-R_j^2} \qquad (8)$$

Values of $VIF$ greater than 5 were associated with a non-acceptable level of multicollinearity, and the corresponding model was discarded (Montgomery et al., 2012).

## 3.2 Results

The equations of the best regression models (built using two to four variables) are reported in Eqs. (9) to (11) for $\bar{h}_1$ and in Eqs. (12) to (14) for $\bar{h}_{24}$. The $R^2_{adj}$ values in Eqs. (9) to (11) are 0.46, 0.52 and 0.54, respectively. The coefficients of determination are higher for $\bar{h}_{24}$, i.e. 0.66, 0.67 and 0.68 (Eqs. (12) to (14), respectively).

$h_1 = 20.3163 - 0.0080 \cdot z + 0.0117 \cdot MAR \qquad (9)$

$h_1 = - 21.6293 - 0.0061 \cdot z + 0.0134 \cdot MAR + 26.2682 \cdot OP \qquad (10)$

$h_1 = - 10.6928 - 0.0051 \cdot z - 0.0273 \cdot C + 0.0131 \cdot MAR + 19.8449 \cdot OP \qquad (11)$

$h_{24} = 16.1392 - 0.0937 \cdot C + 0.0712 \cdot MAR \qquad (12)$

$h_{24} = 33.1529 - 1.8574 \cdot 10^{-5} \cdot LON - 0.1319 \cdot C + 0.0701 \cdot MAR \qquad (13)$

$h_{24} = 127.2773 - 2.9498 \cdot 10^{-5} \cdot LON - 1.9130 \cdot 10^{-5} \cdot LAT - 0.0971 \cdot C + 0.0735 \cdot MAR \qquad (14)$

Considering the three $\bar{h}_1$ models (Eqs. (9) to (11)), it is possible to notice the negative slope coefficient associated with elevation. This confirms what was discussed in the previous section. On the other hand, it is remarkable that the best models found for $\bar{h}_{24}$ do not include the elevation: this outcome can be explained by considering the fact that $MAR$ is always significant, regardless of the number of variables. Models in which $MAR$ was excluded actually present $z$ as a significant covariate, but with less relevance than the regression models for $\bar{h}_1$.

Regardless of the number of the variables considered, and despite the marked increase in the corresponding value of $R^2_{adj}$, compared to the simple regression, we found that the residuals of the multivariate regressions were still characterized by spatial clustering and high local errors, basically all in the same areas in Figure 2. In other words, resorting to additional variables but keeping a uniform relationship between each variable and precipitation over all of Italy does not produce a decisive reduction in the bias for large areas of the country. Thus, we decided to deconstruct the modeling approach and to look for clues of distinct generating mechanisms in distinct areas of Italy.

## 4 Sub-national scale multiple regression analysis

### 4.1 Methods

In this section, an additional paradigm is introduced into the models for the spatial variability of precipitation to reduce the spatial bias, namely the selection of limited areas to build "local" regression models, as an alternative to using data for the whole of Italy. Such an attempt was already made by Caracciolo et al. (2012), who borrowed the subdivision criterion from previous regional frequency analyses. In this work, we have focused on the role of geography and morphology on the spatial variability of annual maximum rainfall depths.

To better understand how to move from national scale relationships to relationships valid for smaller areas, we started by

considering the Alpine area separately from the Apennine region along the entire peninsula, and from the two main islands (Sicily and Sardinia; see Figure 1 for the geographic positions), as a first approximation. We then built four different multivariate models: 1. The Alpine region (i.e., from Piedmont, including the western part of Liguria, eastward up to Friuli Venezia Giulia; this region was delineated using the SOIUSA classification, as suggested by Accorsi, 2016); 2. The Apennine region, including peninsular Italy; 3. Sicily; 4. Sardinia. We evaluated the best regression models for these four regions, as

described in Sect. 3.1, using up to four covariates.

## 4.2 Results

The new set of models built for the four regions were tested by computing the error statistics over the entire country. The obtained results indicated that they provided higher $R^2_{adj}$ than for the national case and better error statistics (see Table 2 for a comparison with the previous multivariate approach). The better results achieved in terms of *RMSE*, *MAE*, *NSE* at the national

scale are due to the improvements obtained for the two main islands (Sicily and Sardinia). More insights are provided in Sect. 6.

| Error statistic | $\bar{h}_1$ – Nation | $\bar{h}_1$ – 4 regions | $\bar{h}_{24}$ – Nation | $\bar{h}_{24}$ – 4 regions |
|:---:|:---:|:---:|:---:|:---:|
| Bias (mm) | 0 | 0 | 0 | 0 |
| MAE (mm) | 3.83 | 3.65 | 13.14 | 11.71 |
| RMSE (mm) | 4.98 | 4.77 | 18.43 | 16.53 |
| NSE (mm) | 0.54 | 0.58 | 0.68 | 0.74 |

**Table 2. Error statistics of the multiple regression models at a national scale and for the 4 macro-regions described in Sect. 4.1, for $\bar{h}_1$ and $\bar{h}_{24}$.**

It is interesting to compare the results obtained for the individual Alpine region with those of Allamano et al. (2009), who analyzed almost the same area. In that case, the ADD curve parameters appeared to be related to elevation and longitude. For the different durations Allamano et al. (2009) also estimated a regression model by linear regression between rainfall depth, elevation and longitude. The dependence of short-duration rainfall on elevation and longitude was found to be statistically significant for all the time intervals, except for the 1-hour duration: in this case, the longitude was not statistically significant.

In our application, the best relationships found for $\bar{h}_1$ and $\bar{h}_{24}$ are those of Eqs. (15) and (16) (characterized by $R^2_{adj} = 0.75$ and 0.76, respectively):

$$\bar{h}_1 = 60.9365 - 1.6664 \cdot 10^{-5} \cdot LAT - 0.0046 \cdot z + 0.0148 \cdot MAR + 25.1825 \cdot OP \tag{15}$$

$$\bar{h}_{24} = 59.0632 - 7.2955 \cdot 10^{-5} \cdot LON - 0.2223 \cdot C + 0.4306 \cdot OBS + 0.0822 \cdot MAR \tag{16}$$

As expected, the $\bar{h}_1$ - $z$ relationship has a negative slope and the Eq. (15) does not include the longitude as covariate, in

agreement with Allamano et al. (2009). The same negative relationship is found in a 24-hour equation that include $z$ (which produces an $R^2_{adj} = 0.74$, that is, lower than that of Eq. (16)). These findings confirm the results of Allamano et al. (2009), who found a general decrease in rainfall depth for increases in elevation for all the durations (up to 24 hours). Eq. (16) also confirms that, although $\bar{h}_1$ decreases systematically with elevation over the whole alpine region, the dependence of $\bar{h}_{24}$ on $z$ decreases as the longitude increases, i.e. moving westward.

The full set of equations used for the four regions are provided in Supplement n°2, together with the $R^2_{adj}$.

Although the improvements achieved with multivariate models over the simple regressions are evident, we found that they were not decisive in providing a homogeneous spatial distribution of the errors. We in fact observed that, even with the best model, we were not able to reduce the clustering effect shown in Figure 2 for the peninsular region (see also Supplement n°3). We believe that a model capable of describing the observed spatial variability of the index rainfall simultaneously at a national

and a local level requires additional insights, which can be obtained using a finer spatial segmentation of Italy.

## 5 Local-scale simple regression analysis of morphological regions

### 5.1 Methods

On the basis of the considerations presented above pertaining to the spatial clustering of residuals, we examined the possibility of obtaining a meaningful segmentation of large areas in subdomains that could be used to obtain "local" relationships between annual maximum rainfall depths and terrain properties. The main reasoning behind the segmentation is that some macroscopic morphological differences can determine markedly different behaviors of the relationships between rainfall and elevation (or other local variables). One example concerns what happens in the windward and leeward sides of mountain ridges, which represent transversal obstacles to the humid masses coming from the sea. Accordingly, we considered some general geomorphological classifications of the landscape that delineate homogeneous areas based on the homogeneity of the macroscopic land properties, such as convexity and texture.

We considered four geomorphological classifications (GC) and denominated them as GC1 to GC4, according to their diversity and success in the geomorphological literature (see the Data Availability section for more information).

The first considered classification, called GC1, was proposed by Iwahashi and Pike (2007); they classified the Earth's surface into 16 topographic types, at a 1-km resolution, based on slope gradient, local convexity and surface texture. We vectorized the raster map, which is available on the European Soil Data Centre website, and then, to reduce the presence of small areas, that could have an extent of just a few km$^2$, all the areas covered by less than 10 pixels (10 km$^2$) were merged with the adjacent class. Among the four different classifications that were used, this is the only one that has a worldwide coverage, as all the other classifications are available at a national scale. A detailed description of the methodologies used by the authors is available in the related references, thus allowing all the classifications to be reproduced over other nations.

The second classification – GC2 – is the "*Carta delle Unità Fisiografiche dei Paesaggi italiani*" ("Map of the physiographic units of Italian landscapes") and is included in the "*Carta della Natura*" ("Map of Nature"; Amadei et al., 2003). A vector maps, which was obtained by means of a visual interpretation of satellite images aided by the analysis of further land cover maps and morphological - lithological characteristics, was available at a 1:250,000 scale.

The third classification – GC3 – was proposed by Guzzetti and Reichenbach (1994). It was obtained, in vector format, by combining an unsupervised three-class cluster analysis of four properties of altitude (altitude itself, slope curvature, frequency of slope reversal and elevation-relief ratio) with a visual interpretation of morphometric maps and an inspection of geological and structural maps.

The fourth classification – GC4 – is the one that delineates areas with the greatest detail, as it is based on local morphometric properties of the landscape. It was proposed by Alvioli et al. (2020), who considered a set of 439 watersheds, covering the whole of Italy, grouped into seven clusters on the basis of the various properties of the slope units within each basin, e.g. a distribution of slope units sizes and aspects. In this work, adjacent watersheds of the same class were collapsed (GIS dissolve), thus producing a total of 178 areas. Geomorphologically homogeneous terrain partitions were defined as "slope units" that were delimited by drainage and divide lines and delineated with a method that was first introduced by Alvioli et al. (2016), and which is widely used in the literature for geomorphological zonation purposes.

An additional geomorphological classification, which was proposed by Meybeck et al. (2001) and which has a worldwide coverage, was also considered. It is based on a combination of a relief roughness index and elevation, and in principle could have been a good fifth candidate. However, it was not included in this analysis because, except for a very large geographical zones, the resulting delineated areas often contained very few rain gauges, which would have made it impossible to perform the desired statistical analyses.

Coherently with the aim of addressing connections between terrain properties and rainfall at a more local level, we built a set of linear regression models between elevation and index rainfall for all the classifications, considering an individual model for each outlined geomorphological zone. Only the internal rain gauges in each of these homogeneous areas with a minimum of

5 available stations that had to ensure at least 100 m of difference in elevation were considered for the regressions. There were four possible outcomes of the applications: a) a positive and significant correlation (at the 5% level); b) a negative and significant correlation; c) a non-significant correlation; d) an insufficient number of stations or an insufficient difference in elevation.

## 5.2 Results

The results for $\bar{h}_1$ are presented hereafter, while the details on $\bar{h}_{24}$ are available in Supplement n°4. The results obtained for each geomorphological zone are mapped in Figures 4(a-d). Blue areas denote geographical zones where $\bar{h}_1$ increases together with the elevation, while the red palette applies to zones where rainfall decreases with the elevation. The color intensity is proportional to the respective slopes. The light gray color denotes zones in which the linear regression is not statistically significant (at a 5% level), while dark gray denotes insufficient data (case d). A comparison of the maps (Figure 4a to Figure 4d) clarifies that the more detailed the geomorphological zonation is, the less likely it is to satisfy the requirements necessary to build a significant regression. On the other hand, if one applies regression models to finer geomorphological classifications, it is possible to see that the regression sign is not uniform over the entire country. For example, with regard to 1-hour data, it is possible to clearly recognize the presence of zones with a positive rainfall depth versus elevation trend for pre-hill/plain morphology in both GC3 (Figure 4c) and GC4 (Figure 4d).

The spatial distribution of the light gray zones is an important information: no trend can be assumed over these areas, because the p-value is greater than 0.05. Consequently, $\bar{h}_1$ can be considered constant over these areas. Finally, the occurrence of the dark gray zones is directly connected to the kind of classification: the smaller the areas delineated by the classification are, the more likely it is that the requirement of having at least 5 rain gauges with at least 100 m difference in elevation is not satisfied. In this regard, we can observe that the above requirements are not met for the elevation difference, i.e. in plain areas, and it is necessary to assume a constant $\bar{h}_1$ in the area as being the most reasonable value.

The maps in Figures 4(a-d) show the availability of more detail in the spatial analysis of the relationship between rainfall depth and elevation has a remarkable effect on both the sign of the regression and the slope of the regression line in several areas. In addition, even the quality of the relationship can improve, as can be seen from a comparison of Figures 4(e-h): far more areas with high $R^2$ can be seen in Figure 4h than in Figure 4e. This allows us to conclude that lower values of $R^2$ are obtained in wider areas.

The same analysis was conducted on $\bar{h}_{24}$, where all of the above outcomes were confirmed, except for the sign of the precipitation vs elevation relationship (Supplement n°4).

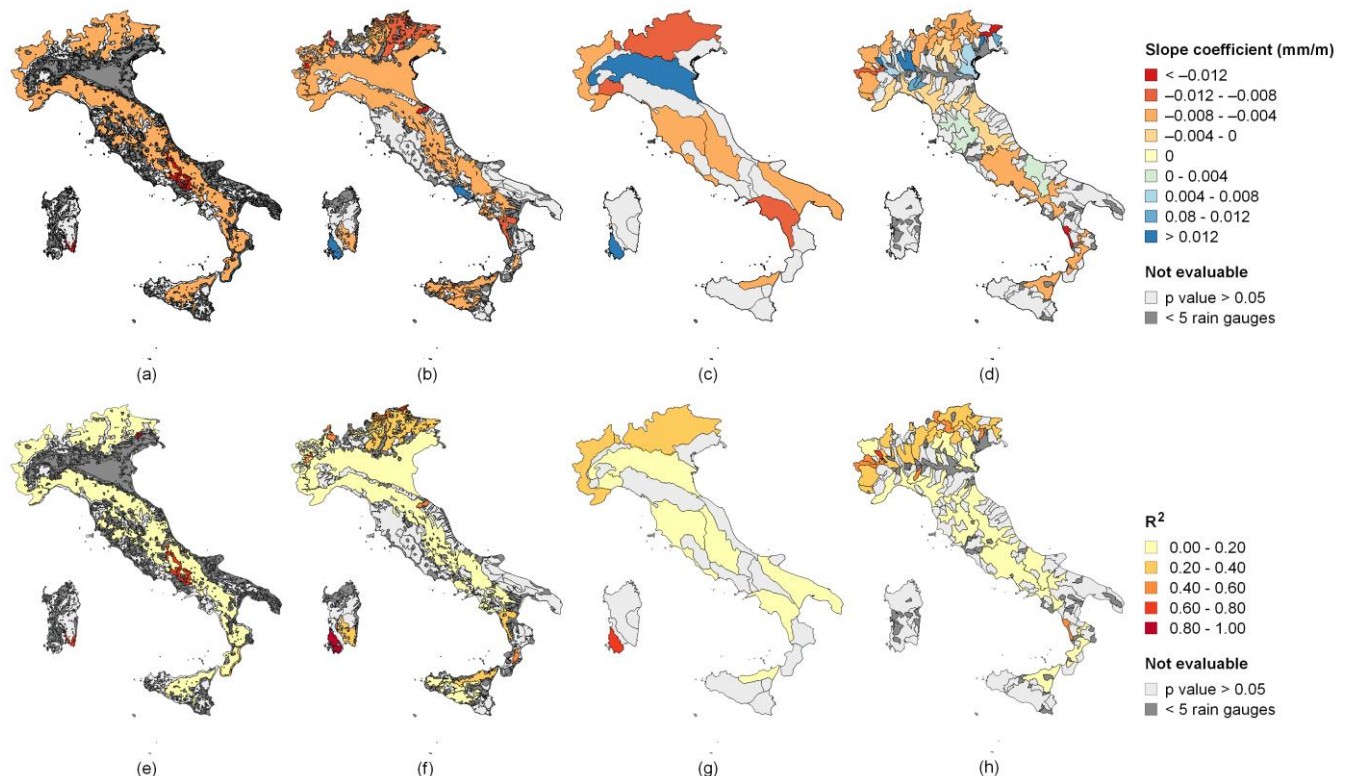

**Figure 4. Slope coefficients of the regression between the mean 1-hour rainfall depth and elevation for GC1 (a), GC2 (b), GC3 (c) and GC4 (d); the $R^2$ of the regression between the mean 1-hour rainfall depth and elevation for GC1 (e), GC2 (f), GC3 (g) and GC4 (h). Geomorphological data source: Iwahashi and Pike (2007), Amadei et al. (2003), Guzzetti and Reichenbach (1994) and Alvioli et al. (2020).**

### 5.3 Error analysis

To test the reliability of the regression models built over the GCs, the linear equations found in each geomorphological zone were applied to all the rain gauge positions, to obtain errors that could be examined at the country scale. The global indices computed for the GC areas in which the regressions were statistically significant are reported in Table 3 for $\bar{h}_1$ and in Table 4 for $\bar{h}_{24}$. These results clearly show a lower performance of the GC1 than the national-scale regression model. On the other hand, the error statistics in Table 3 and Table 4 show that GC4 produces the smallest errors and this geomorphological subdivision therefore presents the best performances. It is possible to understand this result by considering that GC4 uses watershed units, while the other classifications are based on the automatic processing of digital terrain data.

A constant value of the index rainfall computed as the spatial average of $\bar{h}_d$ was adopted over any areas where the morphological regression model was not statistically significant. In this way, it was possible to compute the statistical indexes at the whole country scale. The error statistics obtained with this application are reported in the last rows of Table 3 and Table 4.

| Regression model | Bias (mm) | MAE (mm) | RMSE (mm) | NSE (-) |
|---|---|---|---|---|
| GC1 Iwahashi and Pike | 0 | 5.94 | 7.67 | 0.10 |
| GC2 Carta della Natura | 0 | 5.65 | 7.18 | 0.15 |
| GC3 Guzzetti and Reichenbach | 0 | 5.15 | 6.77 | 0.27 |
| GC4 Alvioli et al. over statistically significant areas | 0 | 4.53 | 5.84 | 0.50 |
| GC4 Alvioli et al. over the entire nation | 0 | 3.87 | 5.12 | 0.52 |

**Table 3. National-scale error statistics for the 1-hour interval. Statistics for GC1, GC2 and GC3 were only evaluated over areas where the regression was statistically significant at a 5% level, while GC4 was tested on both statistically significant areas and over the entire nation, using the mean rainfall, where there was a p-value > 0.05 or where the requirement of at least 5 rain gauges with at least 100 m difference in elevation was not satisfied.**

| Regression model | Bias (mm) | MAE (mm) | RMSE (mm) | NSE (-) |
|---|---|---|---|---|
| GC1 Iwahashi and Pike | 0 | 30.44 | 39.49 | -0.11 |
| GC2 Carta della Natura | 0 | 20.30 | 31.29 | 0.11 |
| GC3 Guzzetti and Reichenbach | 0 | 20.03 | 28.84 | 0.32 |
| GC4 Alvioli et al. over statistically significant areas | 0 | 14.84 | 21.12 | 0.61 |
| GC4 Alvioli et al. over the entire nation | 0 | 14.36 | 20.73 | 0.60 |

**Table 4. National-scale error statistics for the 24-hour interval. Statistics for GC1, GC2 and GC3 were only evaluated over areas where the regression was statistically significant at a 5% level, while GC4 was tested on both statistically significant areas and over the entire nation, using the mean rainfall, where there was a p-value > 0.05 or where the requirement of at least 5 rain gauges with at least 100 m difference in elevation was not satisfied.**

**6 Discussion**

The different regression models used in this work to investigate the role of morphology on the spatial distribution of sub-daily annual maximum rainfall depths produced results deserving some comments. First of all, it must be mentioned that a nationwide multiple regression model that includes morpho-climatic attributes represents a significant step forward with respect to the simple regression model, as the error statistics show. In this approach, working at a national scale and given the elongated shape of the Italian peninsula, geographic location was expected to play a major role in the spatial distribution of

extremes, even though this evidence was not mentioned in similar national-scale analyses (see e.g. Faulkner and Prudhomme, 1998, for the UK, and Avanzi et al., 2015, for Italy). The role of geography progressively weakened while seeking further improvements, in terms of *MAE* and *RMSE*, through the application of distinct multiple regressions to four macro-regions, i.e. the Alps, the Peninsular Italy and the main islands (Supplement n°5).

Our findings show that while the 24-hour index rainfall exhibits a clear overall dependence on the geographic location at a full

national scale (Eq. (14)), the same does not apply to 1-hour extremes (Eq. (11)). In an area with a lesser span in latitude (the Italian Alps), instead, the 1-hour extremes curiously show some dependence on latitude (Eq. (15)). Formetta et al. (2022) followed a similar reasoning, recognizing the role of geography and elevation as they partitioned by longitude and elevation even a small area (the province of Trento) before applying their statistical analyses.

While the multivariate regression can be a good tool to express geographic dependence, and on 24-hour extremes the national

scale helps in drawing some general findings, the residual errors in large clustered areas are still very significant. Therefore, geographic attributes seem not to drive uniformly the variability of rainfall extremes all over Italy, as the high residuals of the multiple regression over these areas do not apparently follow any latitudinal/longitudinal gradient. These findings can derive only from a national-scale analysis.

The better suitability of the application of multiple regressions on four regions is confirmed by the increase of the adjusted

coefficient of determination ($R^2_{adj}$), as reported in Section 3.2 and in Supplement n°2. Moreover, while the national-scale multiple regression model provides high negative residuals over Sardinia and high positive residuals over Sicily, the four-region multiple regression model significantly improves this result (see Supplement n°3-5 for more details). However, similar improvements were not achieved in the peninsular and alpine areas of the country.

The subsequent investigations undertaken in Section 5 descend from the above considerations, i.e. the building of regressions

in morphological regions that are a fraction of the whole area is an attempt to overcome the highlighted lack of regularity in the dependence between rainfall and geography. Among all the considered geomorphological classifications, the selection of rain gauges for the model application is more effective in the case of GC4 (Alvioli et al., 2020), which embeds also hydrographic information. The GC4 model behaves reasonably well for both the 1- and 24-hours durations, compared to the multiple regression models, as far as the national scale is considered. Table 5 summarizes all the previously mentioned

statistics.

| Regression model | Bias (mm) | MAE (mm) | RMSE (mm) | NSE (-) |
|---|---|---|---|---|
| 1h National simple regression | 1.07 | 5.29 | 6.98 | 0.10 |
| 1h National multiple regression | 0 | 3.83 | 4.98 | 0.54 |
| 1h Four-regions multiple regression | 0 | 3.65 | 4.77 | 0.58 |
| 1h GC4 regression | 0 | 3.87 | 5.12 | 0.52 |
| 24h National simple regression | 6.05 | 22.27 | 31.99 | 0.03 |
| 24h National multiple regression | 0 | 13.14 | 18.43 | 0.68 |
| 24h Four-regions multiple regression | 0 | 11.71 | 16.53 | 0.74 |
| 24h GC4 regression | 0 | 14.36 | 20.73 | 0.60 |

**Table 5. Error statistics for the 1- and 24-hour intervals at a national scale. Average spatial values are used for the gray areas in Figure 4d. The bias of the national simple regression is different from zero being evaluated as $bias_d = \frac{1}{n} \cdot \sum h_{avg}(d) - \hat{a} \cdot d^{\hat{n}}$.**

Analyzing the error statistics computed globally at the national scale, it seems that the four-region multiple regression approach is the most precise. However, this is not necessarily true at a local scale. In order to clarify the drawbacks that large-scale regression models can produce, for the 1h case we compared the residuals obtained from the four-region multiple regression model (Figure 5a) in the areas identified by GC4 with the residuals of the GC4 regression model by selecting: 1) the GC4 areas that were statistically significant (Figure 5b) and 2) the entire nation (Figure 5c). The mean rainfall depths were considered over not statistically significant areas (i.e., the gray areas visible in Figure 5b). The GC4 regression models resulted to be statistically significant for $\bar{h}_1$ for the 45% of the Italian area, for a total of 31 different areas, while the GC4 model for $\bar{h}_{24}$ resulted statistically significant in 49% of the area, for a total of 47 different zones (in the Supplementary Material figures dealing with the 24h case are available). From a comparison of the maps in Figure 5, it is possible to note that the multiple regression model has a spatially non-uniform bias while the average bias obtained from the individual models in the zones selected by GC4 is zero all over Italy. Maps of all the other statistics are reported in Supplement n°5 (Figure S4). This outcome is evident also for the 24-hour case (see, for example, the maps of the bias - Figure S5a-e).

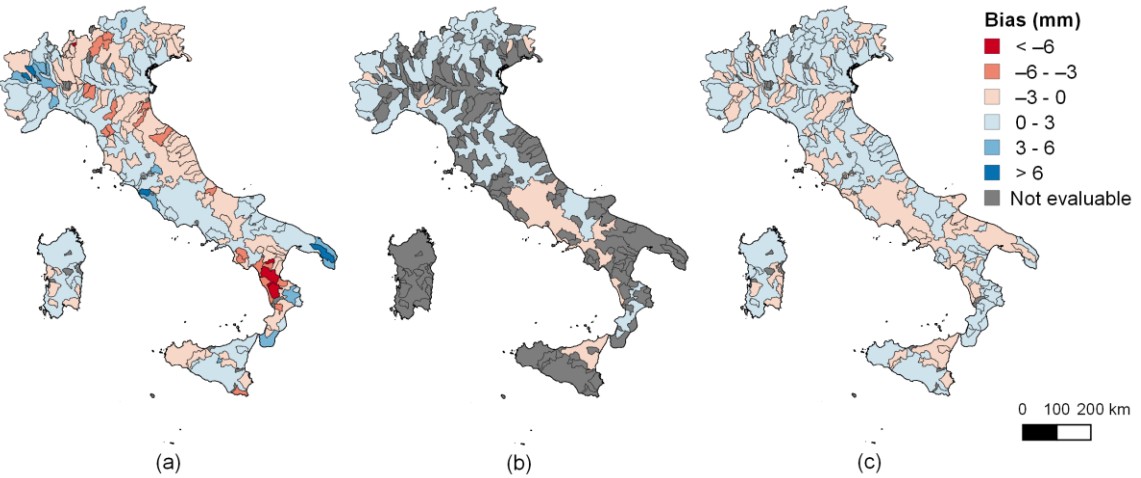

**Figure 5. 1h case. Local bias for the four-region multiple regression model (a), the GC4 simple linear regression model over statistically significant areas (b) and the GC4 simple linear regression model over all the areas (c). Geomorphological data source: Alvioli et al. (2020).**

An additional comparison was undertaken to investigate the local bias. In this case, we computed the bias for each subdomain of GC4. We compared the bias values obtained using the following four conditions: 1) the national-scale simple regression model (Eqs. (3) and (5)), 2) the national-scale multiple linear regression model (Eqs. (11) and (14)), 3) the four-regions multiple regression model (Eqs. (S3), (S6), (S9), (S12), (S15), (S18), (S21) and (S24) in Supplement n°2) and 4) the GC4 simple regression model (Sect. 5).

The results are illustrated in the maps of Figure 6, which shows the best regression model for each area in different colors: Figures 6a-b are related to $\bar{h}_1$, while Figures 6c-d refer to $\bar{h}_{24}$; Figures 6a and 6c only highlight the situations where significant regressions were found. The results in Figures 6b and 6d include the bias calculated in non-statistically significant areas with respect to the spatial average of the rainfall depths. The good results obtained in the areas where the spatial mean values are adopted can be seen by comparing the borders of the GC4 areas with the clusters of the residuals of the multiple regression

model (see Supplement n°3). A dedicated multiple regression model was built for the island of Sardinia: nevertheless, the bias all over the GC4 areas is smaller when the local spatial average is used. A good correspondence between the residual clusters and the GC4 borders is evident.

From Figure 6, it is possible to conclude that the morphological subdivisions allow a set of simple linear regression models to be built that can perform better almost everywhere than the other wide-area models in terms of local bias.

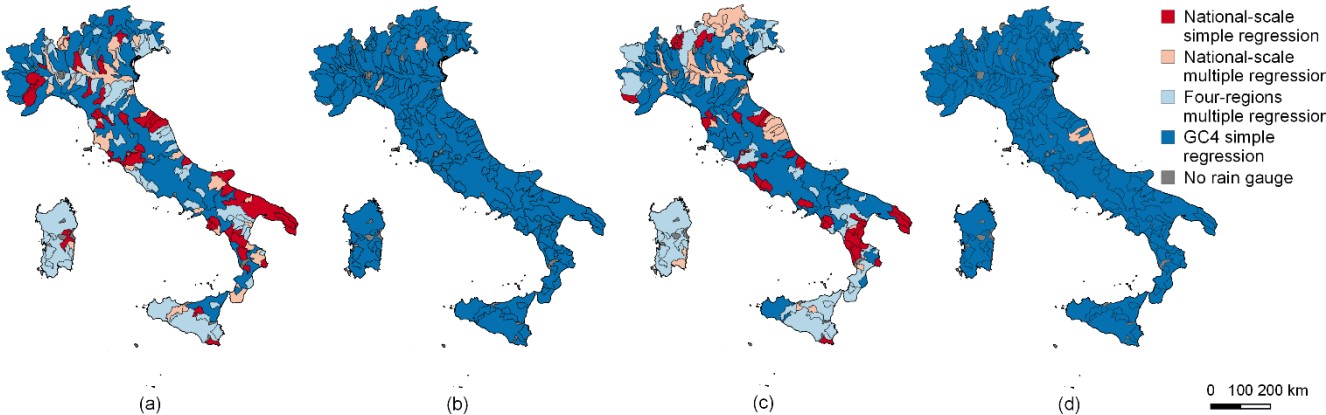


**Figure 6. Absolute bias assessment for all the regression models used for the 1-hour case (a,b) and 24-hour case (c,d). The color refers to the model that provides the lowest absolute value of the bias. The GC4 model bias in cases (a) and (c) was only evaluated for statistically significant areas, while it was evaluated over every area in (b) and (d). Geomorphological data source: Alvioli et al. (2020).**

**7 Conclusions**

In this paper, we have analyzed the role of orography and morphology on short-duration annual maximum rainfall depths, taking advantage of a new and comprehensive database for Italy, I[2]-RED (Mazzoglio et al., 2020). The approach finds its relevance in the first use of the most complete and updated data collection of short-duration annual maxima available for the whole Italian territory. As regards the previous knowledge on the topic, our analyses allowed to better understand, confirm

and extend previous results from the literature.

The results described in this paper show that a national-scale simple regression model of the precipitation vs elevation presents some weaknesses (high residual values, high local- and national-scale bias, low *NSE* coefficient, etc) and therefore needs to be improved.

The use of multiple regression models introduces some benefits, such as a reduction of *MAE* and *RMSE* at the national scale,

nevertheless they were not successful in reducing the local bias.

Considering the necessity of working on smaller domains, we analyzed several geomorphological classifications which are able to preserve the intrinsic value of the statistically significant landscape variables that emerge in regression models. Four different geomorphological classifications available in literature were used to provide criteria for the identification of homogeneous regions. We applied simple linear regression models over these homogeneous domains and compared the

performances at both a national and a local level. Among all the considered classifications, the selection of rain gauges for the model application was found to be more effective in the case of GC4 (Alvioli et al., 2020), which embeds hydrographic information.

The best approach was selected by evaluating the error statistics for the bias at both a national and a local scale, and at a national scale for *MAE*, *RMSE* and *NSE*. The obtained results have shown that using simple linear regression applied to GC4

model performs better than all the others, in the areas in which the GC4 model is statistically significant, in terms of bias. As far as national statistics are concerned, considering the mean rainfall depths in the gray areas in Figure 5b does not significantly affect the performance of GC4, in terms of *MAE*, *RMSE* and *NSE*, in particular for the 1-hour duration. In short, we propose using the GC4 model where possible and adopting the (spatial) mean value of the rainfall depths in case of non-statistically significant relationship.

This work has led to the following conclusions. The relationship between precipitation and elevation is not meaningful in all the areas in Italy, as already pointed out by Caracciolo et al. (2012) for the Island of Sicily. In this work, this concept has systematically been extended to the whole country, and significant relationships have only been obtained for 45% of the area for $\bar{h}_1$ and 49% for $\bar{h}_{24}$. As far as the model that we suggest using is concerned, that is GC4, we are aware that improvements are possible, considering that no significant regressions were found over 55% ($\bar{h}_1$) and 51% ($\bar{h}_{24}$) of the territory. However, it

should be pointed out that the rainfall station density is not sufficient for the application of the here proposed method over 9% of the territory.

Details regarding the model based on GC4 and numerical values of the regression parameters are provided in the Data availability.

**Data availability**

The Iwahashi and Pike geomorphological classification (GC1) is available on https://esdac.jrc.ec.europa.eu/content/global-landform-classification, the "Carta della Natura" classification (GC2) is available on https://www.isprambiente.gov.it/it/servizi/sistema-carta-della-natura, the Guzzetti and Reichenbach classification (GC3) is available upon request to the authors, while the Alvioli et al. classification (GC4) ise available on http://geomorphology.irpi.cnr.it/tools/slope-units.

The rainfall data was obtained from the I$^2$-RED database. Although the Italian law requires an open-source policy for all public data, this right has not yet been implemented by all the Italian agencies involved in the management of the rain gauge network. The agreements we signed with some of these agencies, aimed at monitoring the correct use of the data, restricted their use to the aims of the authors' project. As a result of these legal restrictions, a complete version of I$^2$-RED can only be provided to two groups of people: members of the authors' research group (which is already fully authorized to use the data), and people

who can prove they have received clearance from the regional authorities. The entire quality-controlled database is available on Zenodo (https://doi.org/10.5281/zenodo.4269509), albeit with restricted access. The data can be used by third parties, for an indefinite timeframe, upon having completed an agreement with the authors and with the regional agencies involved in the data collection. The raw data availability depends on the region: a complete description of how to access this data is reported in Mazzoglio et al. (2020).

The model based on GC4 and the numerical values of the regression parameters are available as a Supplement.

**Author contribution**

Conceptualization: PC, IB, PM, MA. Data curation: PM. Formal analysis: PM. Funding acquisition: PC, IB. Investigation: PM. Methodology: PC, IB, PM. Project administration: PC, IB. Resources: PC, IB. Software: PM. Supervision: PC, IB, MA. Validation: PM. Visualization: PM. Writing – original draft preparation: PM. Writing – review & editing: PC, IB, PM, MA.

**Competing interests**

The authors declare that they have no conflict of interest.

**Acknowledgements**

The authors acknowledge the regional agencies involved in the management of the rain gauge networks that provided the rainfall measurements included in I²-RED. Full credits are reported in Mazzoglio et al. (2020).

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
