# Peer review of "The role of morphology on the spatial distribution of short-duration rainfall extremes in Italy"

_Hydrology and Earth System Sciences, 2021_

## Author Comment (AC1)

**Response to reviewer #2 of "The role of morphology on the spatial distribution of short-duration rainfall extremes in Italy" by Mazzoglio et al.**

**C:** The authors here investigate short-duration rainfall extremes in Italy focusing on improving simple relationships with elevation. The form national multiple regression equations exploiting several geomorphological covariates and one climatic. At the national level this did not really led to significantly improvements and thus the authors proceed to several localized spatial clustering approaches identifying this need given the clear spatial pattern in local bias. The paper was joy to read, clearly written, easy to follow and offers an addition to the literature. Therefore, my comments are only minor and optional since the paper fulfills its goal as is.

**R:** *We thank the reviewer for appreciating our work.*

**C:** In general, there are different definition on what is considered extremes. There so many references on extremes, and sub daily extremes, mean extremes, etc. in the introduction but it is not crystal clear what these extremes are. Are POT values, annual/seasonal maxima, etc? Please clarify.

**R:** *We thank the reviewer for this comment. In our work we used annual maximum rainfall depths recorded in different intervals (durations). For the sake of clarity, we will replace the word "extremes" with "annual maximum rainfall depths" or with "index rainfall" when we introduce the average of the annual maxima, according to the literature standards.*

**C:** Section 2.2. Can you show the fitting the Equations 3-6? Especially the nonlinear 5-6.

**R:** *The fitting of the four regression models (Avanzi et al. and this paper) will be introduced in the Supplementary material and is visible in Figure 1 of this document.*

[Figure]

*Figure 1. Fitting of the four equation on $I^2$-RED data.*

**C:** Fig 2 please try to use the minus symbol for minus and not the dash as the "--" is not clear and it should be "- –".

**R:** *We will correct this issue in the new version.*

**C:** Out of curiosity, have the authors tried to see if there's potential in using the temperature as a climatological variable?

**R:** *We have evaluated the possibility of including the mean temperature as an additional covariate in a preliminary step: this step is not documented in the paper, as the temperature comes out highly correlated with the elevation, as expected. In the subsequent analyses we retained only the elevation as an active covariate, considering that the multicollinearity tests were not passed by the equations that contain both the variables temperature and elevation.*

**C:** The authors might be interested in the Moccia et al. 2021 (10.1016/j.ejrh.2021.100906) study (though at daily scale) analyzing a fine resolution gridded product over Italy and investigating extremes. The bivariate choropleth in Fig13 also shows elevation-rainfall depth relationship.

**R:** *We thank the reviewer for this valuable suggestion. This reference will be very useful in a follow-up work we are carrying out, as the suggested paper is based on the use of CHIRPS data (satellite + ground data). In the case of the present paper we intended to focus only on literature work carried out using annual maximum data recorded by rain gauges.*

**C:** Is beta-i a vector in Eq7?

**R:** *Beta is a vector while beta_i is an element of the beta vector. We will correct this in the new version.*

**C:** Was there some preliminary examination in leading to select a multiple linear model? Maybe pair-wise scatter plots would give valuable information and might actually lead in selecting nonlinear relationships at least for some of the covariates. It might worth creating these scatter plots. Also, yes the test showed no collinearity, but let's see this also in scatter plots.

**R:** *In Figure 2 and Figure 3 of this document we reported the scatter plots of the mean of the extremes in 1h and 24h taken in each Italian station, against different covariates. No dominant covariate emerges from these plots: this suggests to proceed towards multiple linear regression models. Their effectiveness is confirmed by the increase of the adjusted coefficient of determination ($R^2_{adj}$) with the increase of the number of covariates (as reported in rows 200-201 and in Supplement n°1).*

[Figure]

*Figure 2. Scatter plots of the mean of the extremes in 1-hour duration against different covariates.*

[Figure]

*Figure 3. Scatter plots of the mean of the extremes in 24-hour duration against different covariates.*

**C:** 210. So in essence MAR acts as a proxy of elevation, right? What is the correlation between MAR and elevation?

**R:** *The correlation between mean annual rainfall (MAR) and elevation is significant, but it is not high enough to be detected by the VIF test. So, equations with both variables are possible. In particular, over the entire nation the correlation coefficient (C.C.) between MAR and elevation is equal to 0.27. If we separately consider the Alps and the rest of Italy we obtain C.C. = -0.24 over the Alps and C.C. = 0.37 over the complementary part. For the whole area of the coastal belt within 20 km from the coastline we obtained the highest correlation, i.e. C.C. = 0.48.*

**C:** Maybe creating subnational regions would be improved by using generic spatial clustering algorithms. I mean based on some statistical properties and not necessarily based on the geomorphological classifications. There are many of them in the literature and could offer an alternative detailed assessment on the optimal number of subregions and on their extend. I hope I have not missed this, but have the authors considered creating such region by applying spatial clustering algorithms to the bias maps?

**R:** *We thank the reviewer for this comment.*

*We have indeed considered the possibility of using spatial clustering algorithms. Drawbacks of these algorithms are that they would create regions with complex shapes (twisted, elongated, etc) that need to be iteratively identified with a high detail level. This is quite difficult if considering that different areas can have very different rain gauge density. Another source of complexity is that a cluster-based approach is based on the definition of some parameters that have to be preset, as the number of clusters and the maximum number of iterations; however, the results obtained are sensitive to the number of clusters and also to outliers (Xu et al., 2015), which makes the preset choices an iterative procedure. Ramos (2001) suggests that it could be useful to use more than one clustering criterion to extract as much information as possible. Bernard et al. (2013) argue that clustering algorithms based on k-means principle is suited when the variable follows a mixture of normal distributions, posing major problems in the analysis of hourly precipitation amounts, that are generally highly skewed. Considering all these degrees of freedom and shortcomings to tackle in a clustering analysis, it would be the matter of an entire new paper but with a quite different approach to undertake as compared to the one proposed here.*

**C:** Summarizing this is a nice paper, adding to the literature and deserves publication.

**R:** *Thank you again for the valuable comments and for having appreciated this work.*

**References:**

*Bernard, E., Naveau P., Vrac M., and Mestre O.: Clustering of maxima: spatial dependencies among heavy rainfall in France, Journal of Climate, 26(20), 7929-7937, https://doi.org/10.1175/JCLI-D-12-00836.1, 2013.*

*Ramos M.C.: Divisive and hierarchical clustering techniques to analyse variability of rainfall distribution patterns in a Mediterranean region, Atmospheric Research, 57(2), 123-138, https://doi.org/10.1016/S0169-8095(01)00065-5, 2001.*

Xu, D. and Tian, Y.: A comprehensive survey of clustering algorithms, Ann. Data. Sci., 2, 165–193, https://doi.org/10.1007/s40745-015-0040-1, 2015.

---

## Author Comment (AC2)

**Response to reviewer #1 of "The role of morphology on the spatial distribution of short-duration rainfall extremes in Italy" by Mazzoglio et al.**

**C:** This study examines the impact of elevation and other geomorphological factors on the distribution of the average annual maximum precipitation of durations of 1 and 24 hours in Italy. The topic is of high scientific interest and relevant to the readers of this journal. The paper has a practical and data-driven cut, and provides quantitative analyses of use to practitioners and local stakeholders. I think it can represent a suitable contribution to this journal, provided some aspects are addressed. The list below only includes references not present in the manuscript. Their purpose is to support the reasoning and should not be considered as recommended.

*R: We thank the reviewer for appreciating our work and for providing useful comments that will allow us to increase the quality of the study. We will review the manuscript according to his suggestion.*

**C:** COMMENT #1. While the paper has a clearly practical cut, I am missing an attempt toward the understanding of the physical reasons behind what is found. I think this should be at least attempted in the discussion, possibly putting the findings in perspective with what is already known on the topic.

*R: Our study was designed to investigate the influence of geomorphological parameters on the spatial distribution of rainfall extremes. We recognize that in our paper the data-based approach is predominant against a physical/analytical approach, but this approach finds its relevance in the first use of the most complete data collection of short-duration annual maxima available for the whole of Italy. As regards the previous knowledge on the topic, our analyses allowed us to better understand, confirm and extend previous results from the literature. They include analyses already available for limited regions in Italy or of other countries (e.g., the one suggested by Prudhomme and Reed for Scotland, mentioned in Section 3.1). In this regard, the discussion section of the paper will be improved by inserting additional comments about the results that we obtained.*

*As an example, we will rewrite rows 241-247 as:*

> *It is interesting to compare the results obtained for the individual Alpine region with those of Allamano et al. (2009), who analyzed almost the same area. In that case, the ADD curve parameters appeared to be related to elevation and longitude. The same study also estimated a regression model by linear regression for the different durations between rainfall depth, elevation and longitude. The dependence of short-duration rainfall on elevation and longitude was found to be statistically significant for all the time intervals, except for the 1-hour duration: in this case, the longitude was not statistically significant. In our application, the best relationships found for $\bar{h}_1$ and $\bar{h}_{24}$ are those of Eqs. (15) and (16) (characterized by $R^2_{adj} = 0.75$ and 0.76, respectively):*

> $$\bar{h}_1 = 60.9365 - 1.6664 \cdot 10^{-5} \cdot LAT - 0.0046 \cdot z + 0.0148 \cdot MAR + 25.1825 \cdot OP \qquad (15)$$

> $$\bar{h}_{24} = 59.0632 - 7.2955 \cdot 10^{-5} \cdot LON - 0.2223 \cdot C + 0.4306 \cdot OBS + 0.0822 \cdot MAR \qquad (16)$$

> *As expected, the $\bar{h}_1$ - z relationship has a negative slope and the Eq. (15) does not include the longitude as covariate, in agreement with Allamano et al. (2009).*

*We will also rewrite the first paragraph of the discussion as:*

*Different regression approaches are used in this work to investigate the role of morphology on the spatial distribution of short-duration annual maximum rainfall depths in Italy. A nationwide multiple regression model that includes morpho-climatic attributes represents a significant step forward with respect to the simple regression model, as the error statistics show. Further improvements, in terms of MAE and RMSE, have been obtained through the application of distinct multiple regressions to four macro-regions, i.e. the Alps, Peninsular Italy and the main islands (Supplement n°4). Their effectiveness is also confirmed by the increase of the adjusted coefficient of determination ($R^2_{adj}$) with the increase of the number of covariates (as reported in Section 3.2 and in Supplement n°1). A national-scale multiple regression model generally provides high negative residuals over Sardinia and high positive residuals over Sicily, while the four-region multiple regression model is able to significantly improve this result (see Supplement n°2-4 for more details).*

*Among all the considered geomorphological classifications, the selection of rain gauges for the model application is more effective in the case of GC4 (Alvioli et al., 2020), which embeds also hydrographic information. The GC4 model behaves reasonably well for both the 1- and 24-hour durations, compared to the multiple regression models, as far as the national scale is considered. Table 5 summarizes all the previously mentioned statistics.*

**C:** COMMENT #2. The state of the art presented in the manuscript mostly focuses on the examined study region (Italy), and is implicitly limited to cover what is relevant to this specific study (i.e. mean annual maxima). However, I feel a wider perspective should be provided, especially when claiming: "The impact of the orography on extreme rainfall depths and […] are still not sufficiently understood for sub-daily rainfall events" (lines 29-30). Specifically, since not much is known, but something has been done, I think that the state of the art should be presented with some more detail. Some examples include Panziera & al 2016, Papalexiou & al 2018, Rossi & al, 2020. Marra & al 2021 is also relevant and currently cited in the manuscript but, to my view, supporting a sentence that is not directly addressed by these authors, so it should be either removed or appropriately presented.

***R:** We agree with the reviewer that discussing the relevance of our manuscript in a wider perspective would help. To provide a broader perspective we will refer to other studies in the Introduction, while keeping the chosen focus on works related to short-duration annual maximum rainfall depths measured with rain gauges. The reference Marra et al. has been removed from row 26. We will insert it in Section 2.2 where it better supports our results.*

**C:** COMMENT #3. One of the main findings presented in the study (lines 14-17) is that elevation alone is as important as (or less important than) geographic location (lat, lon, distance from the coast) in explaining the average annual maximum precipitation, and that the impact of orography emerges more clearly at the local scales (lines 18-20). I have some trouble with this reasoning. I'll try to explain: (a) what the authors present as a finding seems to me rather obvious: geographic location dominates over orography, unless we expect extremes in Caltanissetta (southern Sicily, 568 m a.s.l.) to be more similar to extremes in Aosta (western Alps, 583 m a.s.l., ~1000 km away) than to extremes in Agrigento (southern Sicily, 230 m a.s.l., ~50 km driving from Caltanissetta) due to the similar elevation. I'm not sure how this can be presented as a finding; my underlying hypothesis would be exactly what here is presented as a finding, i.e. that geographic location is more important than elevation, and that elevation modulates the extremes, for example by altering some properties of the storms that occur in a given area; (b) once one accepts the above hypothesis (or finding), the need for "countrywide"

analyses on which the authors put emphasis in the introduction loses part of its scientific interest, while indeed maintaining vast practical interest. To my view, this aspect might require some work on rephrasing introduction and presentation of these results.

*R: We thank the reviewer for raising the issue, that allows us to better clarify the rationale of our approach. Indeed, in a so elongated shape as that of the Italian peninsula we seem to agree that geography should play a major role in the spatial distribution of extremes, even though no such specific was always mentioned at the country level (see e.g. Faulkner and Reed, 1998, for the UK, Avanzi et al., 2015, for Italy). The point is, if this role is found: "how can we quantify" it? How important will be the aggregate information coming from all over a country to justify the findings? Does this piece of information alone carry most of the spatial variance? Let's cover the above points in order.*

*If, on the one hand, the 24-hour index rainfall extremes express (in our findings) a clear overall dependence on the geographic location at a full national scale (Eq. 14), on the other hand the same does not apply to 1-hour extremes (Eq. 11). In an area with a lesser span in latitude (the Italian Alps) the 1-hour extremes curiously show some dependence on latitude (Eq.15). This outcome, once compared with the one at the full national scale, suggests that this relation with 'geography' could derive from some other hidden mechanism. Formetta et al. (2022) seem to follow the same reasoning, accepting empirically a role of geography through the partitioning by longitude of an even smaller area (the province of Trento) before applying their statistical analyses.*

*While the multivariate regression can be a good tool to express geographic dependence -and on 24-hour extremes the national scale helps in drawing some general findings- in our paper we point out that the residual errors in large clustered areas are still very significant, even after considering the role of the geographic location. Therefore, geographic attributes seem not to drive uniformly the variability of rainfall extremes all over Italy, as the high residuals of the multiple regression over these areas do not apparently follow any latitudinal/longitudinal gradient. These findings can only derive from a "countrywide" analysis.*

*The subsequent investigations undertaken in the paper descend from the above considerations, i.e. the building of regressions in morphological regions that are a fraction of the whole area is an attempt to overcome the lack of regularity in the dependence found between rainfall and geography.*

*In the new version we will clarify these points.*

*In order to avoid misunderstanding we will rewrite lines 14/17. We plan to substitute*

> *The results of the national-scale regression analysis did not confirm the assumption of elevation being the sole driver of the variability of rainfall extremes. The longitude, latitude, distance from the coastline, morphological obstructions and mean annual rainfall resulted to be significantly related to the index rainfall, and to play different roles for different durations (1- to 24-hours).*

*with*

> *The results of the national-scale regression analysis did not confirm the assumption of elevation being the sole driver of the variability of the index rainfall. The inclusion of longitude, latitude, distance from the coastline, morphological obstructions and mean annual rainfall contributes to explain a larger percentage of the variance and resulted to be significantly related to the index rainfall and to play different roles for different durations (1- to 24-hours).*

**C:** COMMENT #4. As the authors mention in lines 23-26, orography affects precipitation in rather different ways when one considers the lee side and the wind side of an orographic barrier. This is expected to differently alter the characteristics of extremes at multiple durations, as shown by Avanzi & al. 2015 for the very same region examined in this study, and by other authors in other regions.

I wonder why this information is not included among the factors examined in this study (for example separating typically-wind sides from typically-lee sides), and whether it could represent a relevant explanatory factor in the used models.

*R: We agree with the reviewer on the importance of this topic. In this regard, we can mention the work by Pelosi and Furcolo (2015) who addressed the role of local prevailing wind directions on the wind- and lee-side effects. In their regression-based analysis over the Campania region, they introduced as a covariate the "exposition", defined as the cosine of the angle between the dominant direction of the wet air masses and the principal direction of inertia of each "orographic object". Their approach did not set a final stage to the issue, being not suitable for an automatic extension to the whole of Italy, and for the subjectivity in defining the nature of the "dominant" wind to be computed and used in the analysis.*

*A thorough treatment of this topic was already attempted by Prudhomme and Reed (1998). In their regression models they used a variable named "EXPO" defined on the basis of previous works, like Basist et al. (1994) and Konrad (1996). Basist et al. (1994) defined an exposure variable as the distance between a rain gauge and an upwind barrier whose elevation is at least 500 m higher than the station; Konrad (1996) used a threshold value of 150 m and Prudhomme and Reed used a threshold of 200 m. They concluded that the definition of EXPO has several drawbacks, as it is based on arbitrary thresholds and is defined in a unidirectional way. Thus, they introduced new variables to address these shortcomings, as the distance from the sea, the obstruction and the barrier. Their results confirmed that the obstruction (that is evaluated in the 8 main directions) explains a much larger percentage of variation in the annual maximum daily rainfall than the EXPO variable (evaluated in only one direction).*

*Based on the above considerations, in our work we decided to follow the same approach of Prudhomme and Reed, using barrier, obstruction and distance from the coastline as covariates. On the other hand, we agree that, for its undeniable importance, the interaction between the mountain ridges and the prevailing wind well deserve the massive and specific efforts required to produce concrete results at the Italian scale. It is, however, outside the scopes of this paper and will be hopefully the subject of future works.*

**C:** COMMENT #5. As the authors mention, only few "countrywide" analyses are published, and one of them is indeed in Italy (Avanzi & al 2015) – others are for example Papalexiou & al 2018 and Panziera & al 2016. However, since countries are characterized by vastly different spatial extents, orographic settings and climatic conditions, I suggest using a more general term to define and discuss the examined scale (e.g., continental/regional/local or something on this line, as opposed to "countrywide").

*R: We thank the reviewer for this comment. We will replace the name "countrywide" in the text with "national".*

**C:** COMMENT #6. Lines 97-99: did you check the intermediate durations? If yes, it would be important to mention the results in the text (one sentence is probably enough). Otherwise, one could object that monotonous behaviors between 1 hour to 24 hours were indeed reported in previous studies, but also important deviations from the simple scaling for durations close to 1 hour were found (e.g., Marra & al 2021), asking whether such deviations could imply non-linear or even non-monotonic changes between durations.

*R: In an exploratory systematic analysis we assessed most methods on all the durations. We then decided to continue only on 1 and 24 hours because they are clearly related to different event types (convectives and stratiforms), as stated in rows 97-99. We would prefer not to draw conclusions on the intermediate durations considering that complete tests were not performed. A quite similar analysis is already included in Allamano et al. (2009) that, however, is based on the definition of small areas (delineated in a subjective way) to be used to test the simple-scaling approach. Performing this type of analysis on a high number of small areas is not straightforward, while a national-scale analysis could be, in this case, not so effective. The results of the multiple linear regression analyses in terms of model coefficients could be easily summarized in a table (as performed in Faulkner and Reed, 1998). However, since we decided to discard the multiple regression approach to focus on a local-scale application, we prefer not to put much emphasis on an intermediate method.*

**C:** COMMENT #7. Line 176-177: why did the authors explore a radial limit of 5 km for the computation of the geomorphological factors? Are 5 km sufficient to explain local orographic impacts at multiple durations?

*R: The 5 km radial limit has been obtained after a sensitivity analysis. We tested different distances, from 1 to 30 km. We observed that the correlation coefficient between mean rainfall depth and openness (as function of the radial limit) has an asymptotic horizontal behavior at distances larger than 5 km. This makes this radius measure the most meaningful to use.*

*In the revised version of the manuscript we will include this information.*

Minor:

**C:** Line 11: my understanding is that the study focused on the average annual maxima, perhaps it is better to directly state this instead of "average of rainfall extremes", because extremes are subject to a number of definitions.

*R: We thank the reviewer for having pointed that out. We will correct this definition in the new version.*

**C:** Lines 70-74: it should be perhaps mentioned explicitly that this is the "simple scaling" approximation

*R: We will correct it in the new version.*

**C:** Line 120: "compared to"

*R: We will correct it in the new version.*

**C:** The symbol used for elevation changes between z and Z. I suggest to choose one for consistency.

*R: Thank you for having spotted this inconsistency. We will correct the manuscript using "z".*

**C:** Line 180 and following: usually boldface italic fonts are used to represent vectors (https://www.hydrology-and-earth-system-sciences.net/submission.html).

*R: We will check and correct this part in the next version of the manuscript.*

**C:** Line 186-188: something sounds off with this sentence

*R: We asked to a native speaker of English professional translator to carefully check the language before the new submission.*

**C:** Line 202-207: I wonder whether the equations contain too many significant digits, but I leave this to the authors to assess

*R: If possible, we would prefer to keep the equations as they are.*

**References (Reviewer)**

Panziera, L., Gabella, M., Zanini, S., Hering, A., Germann, U., and Berne, A., 2016. A radar-based regional extreme rainfall analysis to derive the thresholds for a novel automatic alert system in Switzerland, Hydrol. Earth Syst. Sci., 20, 2317–2332, https://doi.org/10.5194/hess-20-2317-2016

Papalexiou, S. M., AghaKouchak, A., & Foufoula-Georgiou, E. (2018). A diagnostic framework for understanding climatology of tails of hourly precipitation extremes in the United States. Water Resources Research, 54(9), 6725–6738. https://doi.org/10.1029/2018WR022732

Rossi, M. W., Anderson, R. S., Anderson, S. P., & Tucker, G. E. (2020). Orographic controls on sub-daily rainfall statistics and flood fre- quency in the Colorado Front Range, USA. Geophysical Research Letters, 47, e2019GL085086. https://doi.org/10.1029/2019GL085086

**References (Mazzoglio et al.)**

Avanzi, F., De Michele, C., Gabriele, S., Ghezzi, A., and Rosso, R.: Orographic signature on extreme precipitation of short durations, J. Hydrometeorol., 16(1), 278-294, https://doi.org/10.1175/JHM-D-14-0063.1, 2015.

Allamano, P., Claps, P., Laio, F. and Thea, C.: A data-based assessment of the dependence of short-duration precipitation on elevation, *Phys. Chem. Earth*, 34, JPCE1649, https://doi.org/10.1016/j.pce.2009.01.001, 2009.

Basist, A., Bell, G.D. and Meentemeyer, V.: Statistical relationships between topography and precipitation patterns, *J. Climate*, 7, 1305–1315, https://doi.org/10.1175/1520-0442(1994)007<1305:SRBTAP>2.0.CO;2, 1994.

Faulkner, D. S. and Prudhomme, C.: Mapping an index of extreme rainfall across the UK, *Hydrol. Earth Syst. Sci.*, 2, 183–194, https://doi.org/10.5194/hess-2-183-1998, 1998.

Formetta, G., Marra, F., Dallan, E., Zaramella, M. and Borga, M.: Differential orographic impact on sub-hourly, hourly, and daily extreme precipitation, *Adv. Water Resour.*, 159, 104085, https://doi.org/10.1016/j.advwatres.2021.104085, 2022.

Konrad, C.: Relationships between precipitation event types and topography in the southern Blue Ridge mountains of the southeastern USA, *Int. J. Climatol.*, 16, 49–62, https://doi.org/10.1002/(SICI)1097-0088(199601)16:1<49::AID-JOC993>3.0.CO;2-D, 1996.

Pelosi, A. and Furcolo, P.: An amplification model for the regional estimation of extreme rainfall within orographic areas in Campania Region (Italy), *Water*, 7(12), 6877-6891, https://doi.org/10.3390/w7126664, 2015.

Prudhomme, C. and Reed, D. W.: Relationships between extreme daily precipitation and topography in a mountainous region: a case study in Scotland, *Int. J. Climatol*., 18(13), 1439–1453, https://doi.org/10.1002/(SICI)1097-0088(19981115)18:13<1439::AID-JOC320>3.0.CO;2-7, 1998.

---

## Author Response (AR1)

**Rebuttal letter of "The role of morphology on the spatial distribution of short-duration rainfall extremes in Italy" by Mazzoglio et al.**

Dear Editor,

we uploaded the revised manuscript titled "The role of morphology on the spatial distribution of short-duration rainfall extremes in Italy".

We revised our manuscript according to the comments received in the Editor report and from the reviewers. A point-by-point reply to all the comments of the reviewers was posted in the open discussion in the previous phase of the revision. In this new phase we do not have new comments to add. Please refer to https://doi.org/10.5194/hess-2021-503-AC2 for the reply on RC1 and to https://doi.org/10.5194/hess-2021-503-AC1 for the reply on RC2.

According to the email received from the editorial support team, we also checked the document for typos, missing co-authors and their affiliations, terminology, updates of data in tables, or updates of variables in equations.

In the following we summarize the modifications that we performed in the revised manuscript.

- We rephrased a few sentences in the abstract to make it more clear, according also to comment n°3 of reviewer n°1.
- Following comment n°2 of reviewer n°2, we replaced (in the abstract and in the manuscript) the word "extremes" with "annual maximum rainfall depths" or with "index rainfall" when we introduce the average of the annual maxima, according to the literature standards.
- Section 1 (Introduction and background) has been expanded including also other relevant studies performed to investigate the spatial variability of rainfall extremes with rain gauge data.
- As requested in comment n°3 of reviewer n°2 we added in Section 2.2 a sentence about the fitting of the four simple regression models. We mentioned that we added them in Supplement S1.
- We followed the comment n°4 of reviewer n°2 and we changed the symbol of the minus that we used in most of the figures: Figures 2, 4, 5, S3, S4 and S5 have been updated.
- Section 3.1 has been expanded to provide additional information about the methods that we used.
- We corrected Equation 7 according to comment n°7 of reviewer n°2 and to the email of the editorial team.
- We replaced $Z$ with $z$ in Equation 9 and following according to the comment received from reviewer n°1.
- Section 4.2 has been expanded in order to compare our results with previous studies.
- While checking the data that we copy-pasted in the tables as requested by the editorial team, we spotted a minor error in Tables 3-4: all the numbers reported in the second column are multiplied by a $10^{-3}$ that we forgot to copy. So, in essence, they are zero, according also to what we reported in Table 2 for the national-scale multiple regression model. In the revised version we corrected this in Table 4 and also in Table 5 (that summarizes the results of Table 2 to 4). In the caption of Table 5 we added a sentence to clarify it.
- Section 6 (i.e. the Discussion) has been expanded, in order to provide some additional comments related to the physical reasons behind what is found.
- In Section 7 we added a sentence to highlight one of the novelties of our approach.

- We also checked the reference section and we updated it by inserting the new articles that we mentioned.
- In the Supplementary material we added the fitting of the four simple regression models as Supplement S1 (as requested in comment n°3 of reviewer n°2).
- Supplement n°2 has been updated replacing $Z$ with $z$.
- In Supplement n°3 we reduced the interval of the colorbar to make the Figures S2a-b more understandable.
- As mentioned before, the figures in Supplement 4 and 5 have been updated to correct the minus symbol. We also corrected the letters of the subplots in the captions.

Best regards,

Paola Mazzoglio on behalf of all the co-authors

---

## Author Response (AR2)

**Rebuttal letter of "The role of morphology on the spatial distribution of short-duration rainfall extremes in Italy" by Mazzoglio et al.**

Dear Editor,

we uploaded the revised manuscript titled "The role of morphology on the spatial distribution of short-duration rainfall extremes in Italy".

We revised our manuscript according to the comment received in the review report. More specifically, we rephrased the sentence of the discussion highlighted by the reviewer.

We also included the credits of the geomorphological boundaries in Figures 4 to 6 following the request received by email from the editorial support team (they asked to check if our figures containing maps/aerial images require a copyright statement/image credit and to add it to the figures, not only in the text).

Best regards,

Paola Mazzoglio on behalf of all the co-authors